# PyLandStats: An open-source Pythonic library to compute landscape metrics

**Martí Bosch** *

Urban and Regional Planning Community (CEAT), École Polytechnique Fédérale de Lausanne (EPFL), Lausanne, Switzerland

* marti.bosch@epfl.ch

**Data Availability Statement:** All the data files are available at a GitHub repository https://github.com/martibosch/pylandstats-notebooks/tree/biorxiv/data and have been derived from the Corine Land Cover datasets https://land.copernicus.eu/pan-european/corine-land-cover.

## Abstract

Quantifying the spatial pattern of landscapes has become a common task of many studies in landscape ecology. Most of the existing software to compute landscape metrics is not well suited to be used in interactive environments such as Jupyter notebooks nor to be included as part of automated computational workflows. This article presents PyLandStats, an open-source Pythonic library to compute landscape metrics within the scientific Python stack. The PyLandStats package provides a set of methods to quantify landscape patterns, such as the analysis of the spatiotemporal patterns of land use/land cover change or zonal analysis. The implementation is based on the prevailing Python libraries for geospatial data analysis in a way that they can be forthwith integrated into complex computational workflows. Notably, the provided methods offer a large variety of options so that users can employ PyLandStats in the way that best supports their needs. The source code is publicly available, and is organized in a modular object-oriented structure that enhances its maintainability and extensibility.

## Introduction

Landscape ecology is based on the notion that the spatial pattern of landscapes strongly influences the ecological processes that occur upon them [1]. From this perspective, quantifying the spatial patterns of landscapes becomes a central prerequisite to the study of the pattern-process relationships. Landscape ecologists often view landscapes as an heterogeneous spatial mosaic of discrete patches, each representing a zone of relatively homogeneous conditions, where the size, shape and configuration of patches significantly affects key ecosystem functions such as biodiversity and fluxes of organisms and materials [2].

Recent decades have seen the development of a series of landscape metrics that quantify several aspects of the spatial pattern of landscapes [3–5]. In a context of significant advances in geographical information systems (GIS) and increasing availability of land use/land cover (LULC) datasets, landscape metrics have been implemented within a variety of software packages [6]. The present article introduces PyLandStats, an open-source library to compute landscape metrics, which represents an advance over previously available software because of its implementation within the most popular libraries of the scientific and data-centric Python stack. Additionally, its modular and object-oriented design allows it to be efficiently used in

**Funding:** The author received no specific funding for this work.

**Competing interests:** The author has declared that no competing interests exist.

interactive environments such as Jupyter notebooks as well as in automated computational workflows, and eases the maintainability and extensibility of the code.

The remainder of the article describes the structure and use of PyLandStats by presenting a thorough example analysis case for a sequence of three raster landscape snapshots of the Canton of Vaud (Switzerland) for the years 2000, 2006 and 2012, which have been extracted from the Corine Land Cover [7] inventory. The code snippets and materials to reproduce the figures of the following four sections can be found in S1, S2, S3 and S4 Codes respectively.

## Analysis of a single landscape

The basic unit of the PyLandStats library is the `Landscape` class, which represents the LULC mosaic of a particular region *at a given point in time*. A `Landscape` instance mainly consists of an array where each position represents the LULC class at the corresponding pixel of the lanscape.

Since LULC data is most often stored in raster files (e.g., GeoTiff), the easiest way to instantiate a `Landscape` object is by passing a path to a raster file as first argument, as in:

```
> ls = Landscape('path/to/raster.tif')
```

The above call will use the rasterio Python library in order to read the raster files, and will extract the pixel resolution and no-data value from the file metadata. Alternatively, `Landscape` instances might also be initialized by passing a NumPy array [8] as first argument, which also requires specifying the x and y coordinates of pixel resolution as a tuple in the `res` keyword argument. By default, PyLandStats assumes that zero values in the array represent pixels with no data. Otherwise, the no-data value can be specified by means of the `nodata` keyword argument. A `Landscape` instance can be plotted at any moment by using its `plot_landscape` method. Note that all the plotting methods of PyLandStats make use of the matplotlib library [9].

## Computing data frames of landscape metrics

Landscape metrics might be classified into two main groups (see S1 Table for the list of metrics implemented in PyLandStats, their classification and their description). The first concerns metrics that provide a scalar value for each patch of the landscape, which are often referred to as patch-level metrics. The second consists of metrics that provide a scalar value that aggregates a characteristic of interest over a set of the patches. This second group allows for an additional distinction between class-level metrics, which are computed over all patches of a given LULC class, and landscape-level metrics, which are those computed over all the patches of a landscape.

For a given `Landscape` instance, the patch-level metrics can be computed by means of the `compute_patch_metrics_df` method as in:

```
# 'ls' is a given 'Landscape' instance
> ls.compute_patch_metrics_df()
```

which will return a pandas data frame [10] as depicted in Table 1, where each row corresponds to a patch of the landscape with its associated LULC class value and the computed metrics.

Similarly, metrics can be computed at the class level by using the `compute_class_metrics_df` method as in:

```
> ls.compute_class_metrics_df()
```

which will return a pandas data frame as depicted in Table 2, where each row corresponds to a LULC class and each column represents a metric computed at the row's class level.

**Table 1. Example data frame of patch-level metrics.**

| patch_id | class_val | area | perimeter | perimeter_area_ratio | shape_index | fractal_dimension | euclidean_nearest_neighbor |
|---|---|---|---|---|---|---|---|
| 0 | 1 | 115.0 | 10600.0 | 92.173913 | 2.409091 | 1.129654 | 1431.782106 |
| 1 | 1 | 13.0 | 2600.0 | 200.000000 | 1.625000 | 1.100096 | 223.606798 |
| 2 | 1 | 2.0 | 600.0 | 300.000000 | 1.000000 | 1.011893 | 223.606798 |
| ⋮ | ⋮ | ⋮ | ⋮ | ⋮ | ⋮ | ⋮ | ⋮ |
| 203 | 2 | 11.0 | 1800.0 | 163.636364 | 1.285714 | 1.052571 | 223.606798 |
| 204 | 2 | 2.0 | 800.0 | 400.000000 | 1.333333 | 1.069990 | 223.606798 |
| 205 | 2 | 14.0 | 2400.0 | 171.428571 | 1.500000 | 1.079705 | 282.842712 |

Lastly, the landscape-level metrics can be computed by using the `compute_landsca-pe_metrics_df` method as in:

```
> ls.compute_landscape_metrics_df()
```

which will return a pandas data frame as depicted in Table 3, where the only row features the values of the metrics computed at the landscape level.

## Customizing the landscape analysis

While a vast collection of metrics have been proposed over the literature of the last decades, many of them are highly correlated with one another. As a matter of fact, Riitters et al. [11] found that the characteristics represented by 55 prevalent landscape metrics could be reduced to only 6 independent factors. Therefore, analysis cases tend to consider a limited subset of metrics. To that end, the three methods that compute data frames of metrics showcased above can be customized by means of the `metrics` keyword argument as in:

```
> ls.compute_class_metrics_df(
    metrics = ['proportion_of_landscape', 'edge_density'])
```

which will return a pandas data frame where only the specified metrics will appear as columns.

On the other hand, certain metrics allow for some customization concerning the way in which they are computed. In PyLandStats, each metric is defined in its dedicated method in the `Landscape` class, which includes metric-specific keyword arguments that allow controlling how the metric is computed. For instance, when computing the edge density (ED), the user might decide whether edges between LULC pixels and no-data pixels (e.g., landscape boundaries) are considered, or whether the area should be converted to hectares. By default, PyLandStats computes the metrics according to the definitions specified in FRAGSTATS v4 [5] (see also S5 Code), and therefore does not consider edges between LULC pixels and no-data pixels, and converts areas to hectares. Nevertheless, the user might decide to change that

**Table 2. Example data frame of class-level metrics.**

| class_val | total_area | proportion_of_landscape | number_of_patches | patch_density | largest_patch_index | total_edge | . . . |
|---|---|---|---|---|---|---|---|
| 1 | 24729 | 7.701939 | 193 | 0.060111 | 2.069921 | 1431600 | . . . |
| 2 | 296346 | 92.298061 | 13 | 0.004049 | 89.451374 | 1431600 | . . . |

**Table 3. Example data frame of landscape-level metrics.**

| | total_area | number_of_patches | patch_density | largest_patch_index | total_edge | edge_density | landscape_shape_index | . . . |
|---|---|---|---|---|---|---|---|---|
| 0 | 321075 | 206 | 0.064159 | 89.451374 | 1431600 | 4.458771 | 9.716931 | . . . |

by providing the `count_boundary` and `hectares` keyword arguments to the `edge_density` method as in:

```
> ls.edge_density()
4.4587713151132915
> ls.edge_density(count_boundary = True)
6.863816865218407
> ls.edge_density(count_boundary = True, hectares = False)
0.0006863816865218407
```

Similarly, the `compute_patch_metrics_df`, `compute_class_metrics_df`, and `compute_landscape_metrics_df` accept the `metrics_kws` keyword argument in the form of a dictionary, which allows setting the keyword arguments that must be passed to each metrics' method when computing the data frames. For instance, in order to compute a class-level data frame with the `proportion_of_landscape` as a fraction instead of a percentage, and include the landscape boundaries in `edge_density`, the `metrics_kws` keyword argument must be provided as in:

```
> ls.compute_class_metrics_df(
    metrics_kws={
      'proportion_of_landscape': {'percent': False},
      'edge_density': {'count_boundary': True}
  })
```

In the above example, the columns of the returned data frame will feature not only the proportion of landscape and edge density, but all the available metrics instead. In order to compute a reduced set of metrics, some of which with non-default arguments, both `metrics` and `metric_kws` keyword arguments must be defined. For instance, in the code snippet below:

```
> ls.compute_class_metrics_df(
    metrics = [
      'proportion_of_landscape', 'edge_density', 'fractal_dimension_am'
    ],
    metrics_kws={
      'proportion_of_landscape': {'percent': False},
      'edge_density': {'count_boundary': True}
  })
```

the returned data frame will be of the form depicted in Table 4.

Note that the `metrics` and `metric_kws` keyword arguments work in the same way for the `compute_patch_metrics_df` and `compute_landscape_metrics_df` methods. Additionally, a list of LULC class values might be provided to the `classes` keyword argument of `compute_class_metrics_df` in order to compute the metrics for the specified subset of classes only. The three keyword arguments are complimentary and might therefore be used in conjunction. For instance, adding a `classes = [1]` to the foregoing code snippet would return a data frame of the form depicted in Table 4 but featuring only the first row.

## Spatiotemporal analysis

Landscape metrics are often applied to assess the spatiotemporal patterns of LULC change for a given region by computing landscape metrics over a temporally-ordered sequence of

**Table 4. Example of a data frame of class-level metrics computed with custom `metrics` and `metrics_kws` keyword arguments.**

| class_val | proportion_of_landscape | edge_density | fractal_dimension_am |
|---|---|---|---|
| 1 | 0.077019 | 4.502998 | 1.129561 |
| 2 | 0.922981 | 6.819590 | 1.204003 |

**Table 5. Example data frame of class-level metrics for a spatiotemporal analysis.**

| class_val | dates | total_area | proportion_of_landscape | number_of_patches | patch_density | largest_patch_index | total_edge | ... |
|---|---|---|---|---|---|---|---|---|
| 1 | 2000 | 24729 | 7.70194 | 193 | 0.0601106 | 2.06992 | 1.4316e+06 | ... |
| | 2006 | 24599 | 7.66145 | 200 | 0.0622907 | 2.02227 | 1.436e+06 | ... |
| | 2012 | 24766 | 7.71346 | 201 | 0.0626022 | 2.02227 | 1.4459e+06 | ... |
| 2 | 2000 | 296346 | 92.2981 | 13 | 0.0040489 | 89.4514 | 1.4316e+06 | ... |
| | 2006 | 296476 | 92.3386 | 8 | 0.00249163 | 89.1318 | 1.436e+06 | ... |
| | 2012 | 296309 | 92.2865 | 8 | 0.00249163 | 89.0916 | 1.4459e+06 | ... |

landscape snapshots. To this end, PyLandStats features the `SpatioTemporalAnalysis` class, which is instantiated with a temporally-ordered sequence of landscape snapshots.

```
> input_filepaths = [
    'snapshot00.tif', 'snapshot06.tif', 'snapshot12.tif'
    ]
> dates = [2000, 2006, 2012] # the dates of each snapshot
> sta = pls.SpatioTemporalAnalysis(input_filepaths, dates = dates)
```

When initializing a `SpatioTemporalAnalysis` instance, a `Landscape` instance will be created for each of the landscape snapshots provided as first argument. The `dates` argument might also be provided as string or `datetime` objects (see S2 Code).

## Computing spatiotemporal data frames

Similarly to `Landscape` instances, the data frames of class and landscape-level metrics of a `SpatioTemporalAnalysis` instance can be computed by means of the `compute_class_metrics_df` and `compute_landscape_metrics_df` methods respectively. For instance, following the snippet above, the data frame of class-level metrics can be obtained as in:

```
> sta.compute_class_metrics_df()
```

which will return a data frame indexed by both the class value and date, as depicted in Table 5.

Similarly, the data frame of landscape metrics can be obtained as follows:

```
> sta.compute_landscape_metrics_df()
```

where the resulting data frame will be indexed by the dates as depicted in Table 6.

Note that PyLandStats does not compute data frames for spatiotemporal analyses at the patch level, given that new patches emerge and others disappear over the years and therefore there is no common index upon which the data frames of patch-level metrics for different snapshots could be assembled.

## Customizing the spatiotemporal analysis

As with the `Landscape` class, the `compute_class_metrics_df` and `compute_landscape_metrics_df` methods of the `SpatioTemporalAnalysis` class also allow customizing how each metric is computed by means of the `metrics` and

**Table 6. Example data frame of landscape-level metrics for a spatiotemporal analysis.**

| dates | total_area | number_of_patches | patch_density | largest_patch_index | total_edge | edge_density | landscape_shape_index | ... |
|---|---|---|---|---|---|---|---|---|
| 2000 | 321075 | 206 | 0.0641595 | 89.4514 | 1.4316e+06 | 4.45877 | 9.71693 | ... |
| 2006 | 321075 | 208 | 0.0647824 | 89.1318 | 1.436e+06 | 4.47248 | 9.73633 | ... |
| 2012 | 321075 | 209 | 0.0650938 | 89.0916 | 1.4459e+06 | 4.50331 | 9.77998 | ... |

**Table 7. Example of a data frame of class-level metrics for a spatiotemporal analysis computed with custom `classes`, `metrics` and `metrics_kws` keyword arguments.**

| class_val | dates | edge_density | fractal_dimension_am | landscape_shape_index | proportion_of_landscape |
|---|---|---|---|---|---|
| 1 | 2000 | 4.503 | 1.12956 | 22.9492 | 0.0770194 |
| | 2006 | 4.51608 | 1.12336 | 23.0892 | 0.0766145 |
| | 2012 | 4.54847 | 1.12347 | 23.181 | 0.0771346 |

`metric_kws` arguments. Additionally, the `classes` keyword argument might be provided to `compute_class_metrics_df` in order to compute the metrics for the specified subset of classes only. For instance, the code snippet below:

```
> sta.compute_class_metrics_df(
    metrics = ['proportion_of_landscape', 'edge_density',
      'fractal_dimension_am', 'landscape_shape_index',
      'shannon_diversity_index'],
    classes = [1],
    metrics_kws = {
      'proportion_of_landscape': {'percent': False},
      'edge_density': {'count_boundary': True}})
```

will return a data frame of the form depicted in Table 7.

Note that although provided within the `metrics` keyword argument, the Shannon's diversity index does not appear in the data frame of Table 7 since it can only be computed at the landscape level. Analogously, the proportion of landscape would not appear in the data frame of landscape-level metrics.

## Plotting the evolution of metrics

One of the most important features of the `SpatioTemporalAnalysis` class is plotting the evolution of the metrics. To that end, the class features the `plot_metric` method, which takes the snake case label of the respective metric name as first argument, e.g., for proportion of landscape, the argument becomes `'proportion_of_landscape'` (see S1 Table for the list of metrics implemented in PyLandStats and their respective snake case labels). In order to plot the evolution of a metric at the class level, the value of the LULC class must be passed to the `class_val` keyword argument as in:

```
# a class value of 1 represents "urban" LULC in this example
> sta.plot_metric('proportion_of_landscape', class_val = 1)
```

which will produce a plot for the metric at the class level as depicted in Fig 1.

If the `class_val` keyword argument is ommited, the metric will instead be plotted at the landscape level. For instance, the following snippet will plot both the class and landscape-level area-weighted fractal dimension in the same matplotlib axis:

```
> ax = sta.plot_metric('fractal_dimension_am', class_val = 1,
                  plot_kws={'label': 'class level (urban)'})
> sta.plot metric(
    'fractal_dimension_am', ax = ax, plot_kws={'label': 'landscape
level'})
> ax.legend()
```

producing a plot as depicted in Fig 2.

In order to customize the resulting plot, the `plot_metric` method accepts, among other keyword arguments, a `plt_kws` keyword argument that will be forwarded to the matplotlib's plot method (see the chapter 2 "Spatiotemporal analysis" of S1 Text).

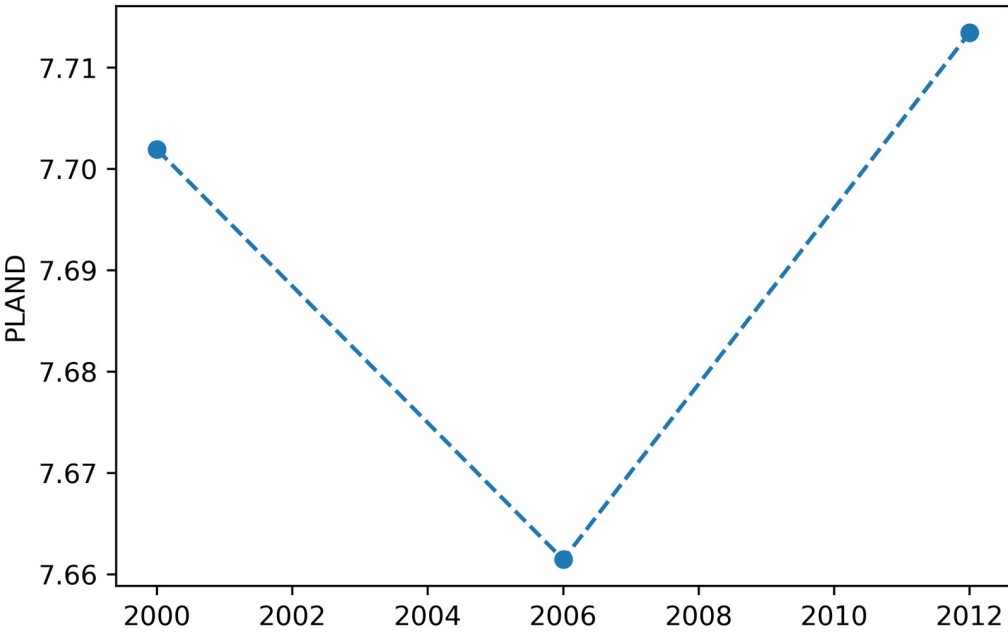

**Fig 1. Example of a plot for a class-level metric in a spatiotemporal analysis.**

## Zonal analysis

Landscape metrics are very sensitive to scale, that is, to the pixel resolution and especially to the spatial extent of the considered map [1, 12, 13]. To overcome such shortcoming, landscape ecologists often turn to methods of multiscale analysis which explicitly consider multiple scales, both in terms of resolution and map extents [14].

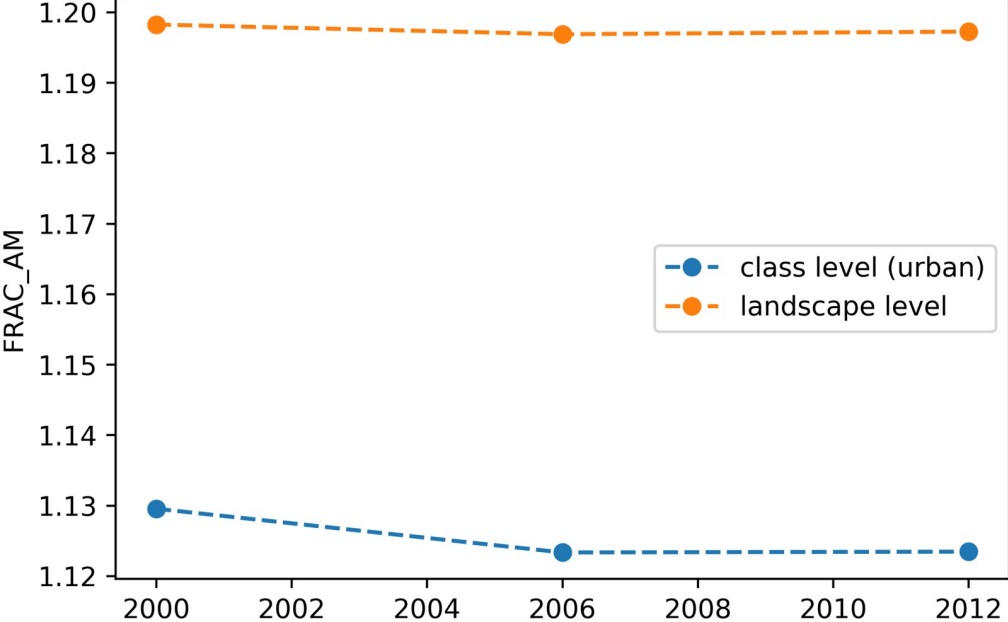

**Fig 2. Example with a metric plotted at both the class and landscape level in a spatiotemporal analysis.**

The PyLandStats library features two classes that might be used for such purpose. The first is `BufferAnalysis`, which segments a given landscape based on a series of buffers of increasing distances around a feature of interest, whereas the more generic `ZonalAnalysis` allows the user to freely choose how the landscape is segmented by providing a list of NumPy masks.

### Buffer analysis around a feature of interest

In line with the classic concentric models of location and land use, evaluating the spatial variation of the environmental characteristics across the urban-rural gradient has become one of the central topics of landscape ecology [15].

Consider a LULC raster file featuring a city and its rural hinterlands. Then, given a coordinate that represents the center of the feature of interest (e.g., a Shapely point with its coordinate reference system) and a list of buffer distances (in meters), a `BufferAnalysis` can be instantiated as follows:

```
> from shapely.geometry import Point
# latitude and longitude of the center of Lausanne in the
OpenStreetMap
> base_mask = Point(6.6327025, 46.5218269)
> base_mask_crs = '+proj = longlat +ellps = WGS84 +datum = WGS84 +no
defs'
# buffer distances (in meters)
> buffer_dists = [10000, 15000, 20000]
# instantiation of 'BufferAnalysis'
> ba = pls.BufferAnalysis(
    path_to_raster, base_mask, buffer_dists,
base_mask_crs = base_mask_crs)
```

where the `BufferAnalysis` instance will generate the landscape of interest for each buffer distance by masking the pixels of the input raster, as illustrated in Fig 3.

On the other hand, the `base_mask` argument might also be a polygon geometry (e.g., administrative boundaries) instead of a point. In such case, note that the list of buffer distances might start from zero in order to start computing the metrics for the region defined by the polygon geometry itself.

Like in the other classes, the data frames of class and landscape-level metrics can be obtained through the `compute_class_metrics_df` and `compute_landscape_metrics_df` methods respectively. For instance, the following snippet:

```
> ba.compute_class_metrics_df()
```

will return a data frame indexed by both the class value and buffer distance, as depicted in Table 8.

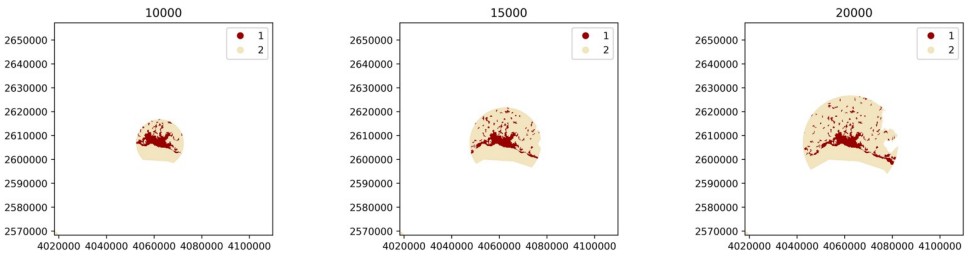

**Fig 3. Landscapes generated by instantiating a `BufferAnalysis` with a raster of urban and non-urban LULC classes (values of 1 and 2 respectively), the coordinates of the city center as base mask, and buffer distances of 10000, 15000 and 20000m (corresponding to the three subplots from left to right).**

**Table 8. Example data frame of class-level metrics for a buffer analysis.**

| class_val | buffer_dist | total_area | proportion_of_landscape | number_of_patches | patch_density | largest_patch_index | total_edge | ... |
|---|---|---|---|---|---|---|---|---|
| 1 | 10000 | 7261 | 24.9648 | 20 | 0.068764 | 21.5472 | 223900 | ... |
| | 15000 | 9630 | 16.7106 | 46 | 0.0798223 | 11.5326 | 395200 | ... |
| | 20000 | 12149 | 13.3476 | 76 | 0.0834981 | 7.30169 | 565200 | ... |
| 2 | 10000 | 21824 | 75.0352 | 4 | 0.0137528 | 74.3614 | 223900 | ... |
| | 15000 | 47998 | 83.2894 | 4 | 0.00694107 | 82.9493 | 395200 | ... |
| | 20000 | 78871 | 86.6524 | 5 | 0.0054933 | 86.3151 | 565200 | ... |

Again, the metrics that are considered in the analysis and how they metrics are computed can be customized by providing the `metrics` and `metrics_kws` keyword arguments respectively to the `compute_class_metrics_df` and `compute_landscape_metrics_df` methods, while the considered classes can be set as the `classes` keyword argument of `compute_class_metrics_df`.

On the other hand, and analogously to the `SpatioTemporalAnalysis` class, the metrics computed for each buffer distance in a `BufferAnalysis` instance can be plotted by means of the `plot_metric` method. Again, `plot_metric` takes an optional `class_-val` keyword argument that if provided, plots the metric at the class level, and otherwise, plots the metric at the landscape level. For instance, the following snippet:

```
> ba.plot_metric('proportion_of_landscape', class_val = 1)
```

will produce a plot for the metric at the class level as depicted in Fig 4.

Another approach to examine how landscape patterns change across the urban-rural gradient is to compute the metrics for each buffer ring that defined between each pair of distances. For instance, for the buffer distances considered in latter example, i.e., 10000, 15000 and 20000, the metrics would be computed for the buffer rings that go from 0 to 10000 m, 10000-

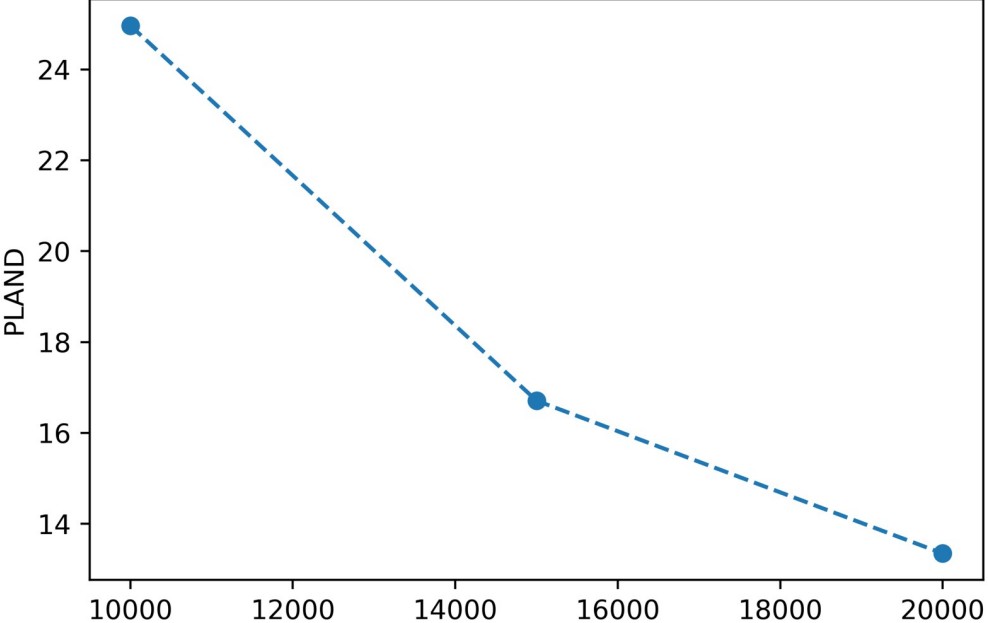

**Fig 4. Example of a plot for a class-level metric in a buffer analysis.** The x axis corresponds to the buffer distances.

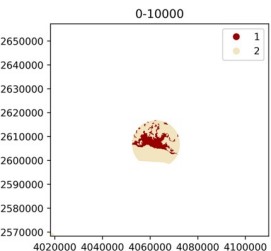 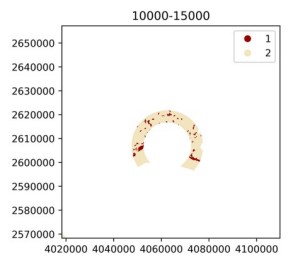 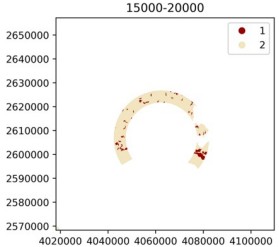

**Fig 5. Landscapes generated by instantiating a `BufferAnalysis` with a raster of urban and non-urban LULC classes (values of 1 and 2 respectively), the coordinates of the city center as base mask, and buffer distances of 10000, 15000 and 20000m (corresponding to the three subplots from left to right) and `buffer_rings` set to `True`.**

**Table 9. Example data frame of class-level metrics for a buffer analysis computing the metrics for the buffer rings.**

| class_val | buffer_dist | total_area | proportion_of_landscape | number_of_patches | patch_density | largest_patch_index | total_edge | ... |
|---|---|---|---|---|---|---|---|---|
| 1 | 0-10000 | 7261 | 24.9648 | 20 | 0.068764 | 21.5472 | 223900 | ... |
| | 10000-15000 | 2369 | 8.29976 | 37 | 0.129629 | 1.68518 | 168600 | ... |
| | 15000-20000 | 2519 | 7.54372 | 37 | 0.110805 | 3.11152 | 169100 | ... |
| 2 | 0-10000 | 21824 | 75.0352 | 4 | 0.0137528 | 74.3614 | 223900 | ... |
| | 10000-15000 | 26174 | 91.7002 | 3 | 0.0105105 | 83.6282 | 168600 | ... |
| | 15000-20000 | 30873 | 92.4563 | 8 | 0.0239578 | 76.117 | 169100 | ... |

15000 m and 15000-20000 m. Such analysis can be performed in PyLandStats by setting the keyword argument `buffer_rings` to `True`, as in the snippet below:

```
> ba = pls.BufferAnalysis(
    input_filepath, base_mask, buffer_dists,
base_mask_crs = base_mask_crs,
    buffer_rings = True)
```

where `BufferAnalysis` will generate the landscapes as depicted in Fig 5.

Under such circumstances, the buffer distance of each in the data frame of class and landscape-level metrics will be strings that represent the buffer distances that correspond to the start and end of each ring, as depicted in Table 9.

Accordingly, the `plot_metric` method of a `BufferAnalysis` will produce a figure as depicted in Fig 6, where the x axis represents the buffer distances of the rings.

## Generic zonal analysis

In certain analysis cases, the user might consider more appropriate to compute the metrics along a decomoposition of the landscape different than concentric buffers, for example, rectangular transects. To that end, PyLandStats features the `ZonalAnalysis` class, which instead of a base mask, accepts a list of boolean arrays of the same shape of our landscape as masks to define our transects (or any other type of subregion). Consider the code snippet below:

```
# this reads the input raster landscape and creates a boolean base
mask
# of the same shape of the landscape and filled with 'False' values
with rasterio.open(input_filepath) as src:
    base_mask_arr = np.full(src.shape, False)

masks_arr = []
```

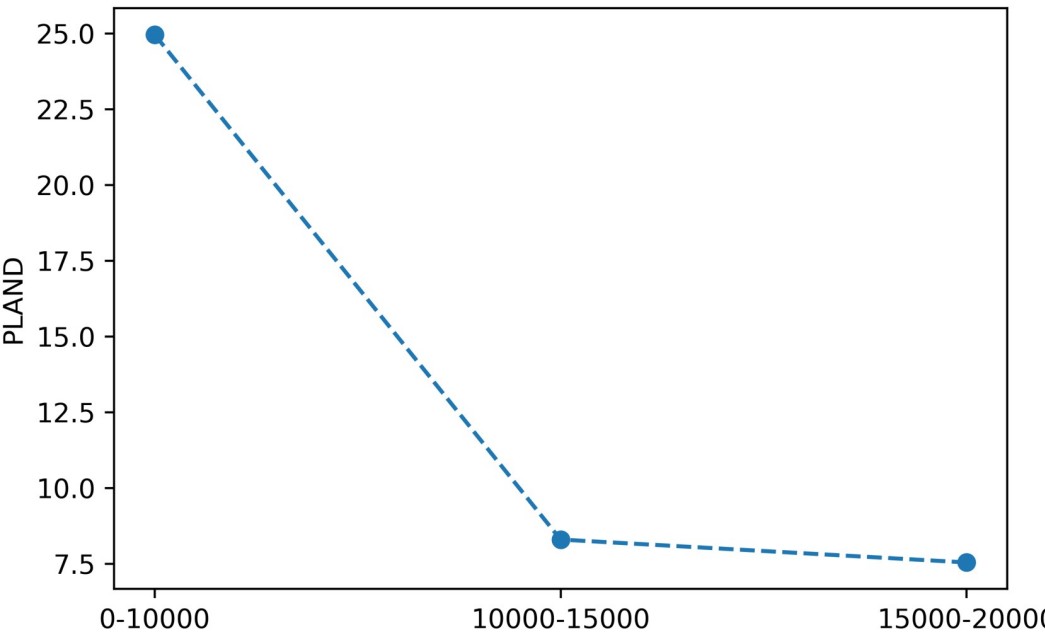

**Fig 6. Example of a plot for a class-level metric in a buffer analysis that computes the metrics for the buffer rings.** The x axis delineates three discrete points, each corresponding to a buffer ring, and whose label represents the ring's start and end buffer distance.

```
# for a pixel resolution of 100m, this corresponds to transects of
30km
transect_len = 300
# this will iterate over three transects (0-30km, 30-60km, 60-90km)
for transect_start in range(0, 900, transect_len):
    mask_arr = np.copy(base_mask_arr)
    # the 400 and 600 serve to slice the landscape vertically along the
    # 20km where the feature of interest is located
    mask_arr[400:600,transect_start:transect_start+transect_len] =
True
    masks_arr.append(mask_arr)
```

where the variable `masks_arr` will be a list of three NumPy boolean arrays, each corresponding to a distinct rectangular transect, as plotted in Fig 7.

The instantiation of `ZonalAnalysis` requires the list of mask arrays (e.g., the `masks_arr` variable created above) as second argument. Additionally, the keyword argument `attribute_values` might be used to map an identifying value or label to each of our landscapes. In this example, a list of strings will be provided in a form which denotes that each landscape corresponds to the transect from kilometers 0 to 30, 30 to 60 and 60 to 90 respectively:

```
> attribute_values = ['0-30', '30-60', '60-90']
> za = pls.ZonalAnalysis(
    input_filepath, masks_arr, attribute_values = attribute_values)
```

where `ZonalAnalysis` will generate the landscapes as depicted in Fig 8.

In `ZonalAnalysis` instances, the data frames of metrics are indexed by the values provided to the keyword argument `attribute_values` as depicted in Table 10.

Again, the data frames of metrics `ZonalAnalysis` can also customized by providing the `metrics` and `metrics_kws` keyword arguments to the `compute_class_metrics_df`

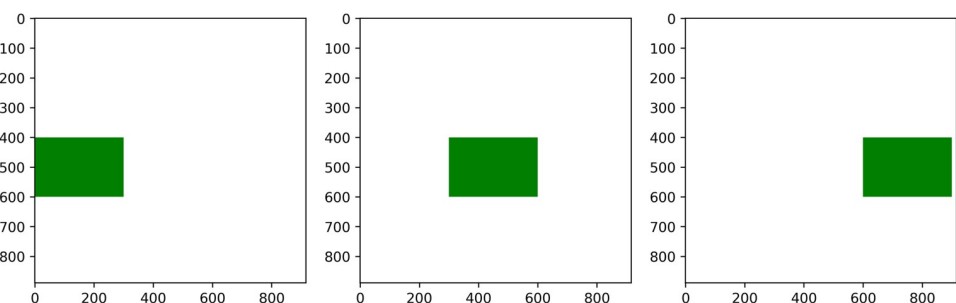

**Fig 7. Example of a list of three boolean mask arrays that delineate three rectangular transects of a landscape.**

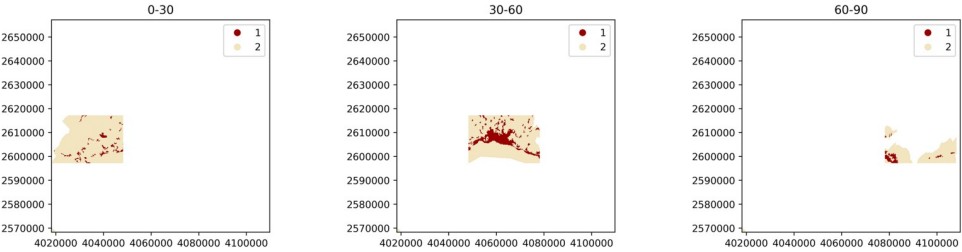

**Fig 8. Landscapes generated by instantiating a `ZonalAnalysis` for three rectangular transects.**

and `compute_landscape_metrics_df` methods, and additionally by the `classes` keyword argument in `compute_class_metrics_df`.

In order to plot a metric's computed value for each subregion, the class `ZonalAnalysis` features a `plot_metric` method which works in the same way as its counterpart in `SpatioTemporalAnalysis` and `BufferAnalysis`. For instance, the following snippet:

```
> za.plot_metric('proportion_of_landscape', class_val = 1)
```

will produce a plot for the metric at the class level as depicted in Fig 9.

## Spatiotemporal buffer analysis

The zonal analysis methods presented above are themselves multiscale analysis approaches since they explicitly consider multiple map extents. Accordingly, the `BufferAnalysis` and `ZonalAnalysis` classes might be employed to obtain scalograms, namely, response curves of the metrics to changing the map extent [16].

Nevertheless, when performing spatiotemporal analyses, it might also be useful to evaluate how the computed time series of metrics responds to changes in the map extent. To that end,

**Table 10. Example data frame of class-level metrics in a zonal analysis of three transects.**

| class_val | attribute_values | total_area | proportion_of_landscape | number_of_patches | patch_density | largest_patch_index | total_edge | . . . |
|---|---|---|---|---|---|---|---|---|
| 1 | 0-30 | 2641 | 5.0768 | 37 | 0.0711251 | 0.707407 | 216700 | . . . |
| | 30-60 | 9577 | 17.6965 | 40 | 0.0739126 | 12.2806 | 370500 | . . . |
| | 60-90 | 1761 | 9.27281 | 9 | 0.0473909 | 6.90854 | 71900 | . . . |
| 2 | 0-30 | 49380 | 94.9232 | 2 | 0.0038446 | 94.9194 | 216700 | . . . |
| | 30-60 | 44541 | 82.3035 | 6 | 0.0110869 | 81.8859 | 370500 | . . . |
| | 60-90 | 17230 | 90.7272 | 6 | 0.0315939 | 53.2199 | 71900 | . . . |

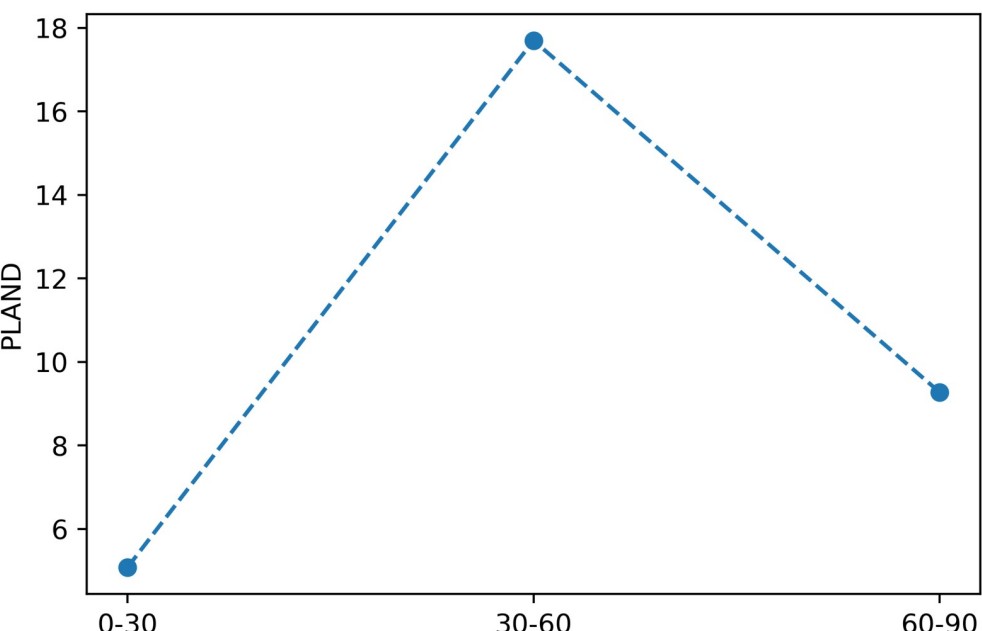

**Fig 9. Example of a plot for a class-level metric in a zonal analysis of three transects.** The x axis corresponds to the values provided to the keyword argument `attribute_values` provided to the initialization of `ZonalAnalysis`.

PyLandStats features an additional `SpatioTemporalBufferAnalysis` class, which is instantiated like a `BufferAnalysis` except that the first argument is a temporally-ordered list of landscape raster snapshots—like in the `SpatioTemporalAnalysis` class—instead of a single raster landscape. In addition, like the `SpatioTemporalAnalysis` class, a list with the dates that correspond to each of the landscape snapshots can be passed to the keyword argument `dates`. Putting it all together, `SpatioTemporalBufferAnalysis` can be instantiated as in:

```
# Note: 'input_filepaths' is a list (like in 'SpatioTemporalAnalysis')
> stba = pls.SpatioTemporalBufferAnalysis(
    input_filepaths, base_mask, buffer_dists,
    base_mask_crs = base_mask_crs, dates = [2000, 2006, 2012])
```

Like `BufferAnalysis`, a `SpatioTemporalBufferAnalysis` can also be instantiated from a polygon geometry. The data frame of class and landscape-level metrics can be computed by means of the the `compute_class_metrics_df` and `compute_landscape_metrics_df` methods respectively, which again, might also be customized by providing the `metrics` and `metrics_kws` keyword arguments, and additionally by the `classes` keyword argument in `compute_class_metrics_df`. In `SpatioTemporalBufferAnalysis` instances, the data frames are indexed by the buffer distances and the snapshot dates (and also by the LULC class values in the class-level data frame, as depicted in Table 11).

The `SpatioTemporalBufferAnalysis` class features a `plot_metric` method with the same signature of its counterparts in `SpatioTemporalAnalysis`, `BufferAnalysis` and `ZonalAnalysis`. For example, the snippet below:

```
> stba.plot_metric('fractal_dimension_am')
```

**Table 11. Example data frame of class-level metrics in a spatiotemporal buffer analysis.**

| buffer_dist | class_val | dates | total_area | proportion_of_landscape | number_of_patches | patch_density | largest_patch_index | total_edge | ... |
|---|---|---|---|---|---|---|---|---|---|
| 10000 | 1 | 2000 | 7261 | 24.9648 | 20 | 0.068764 | 21.5472 | 223900 | ... |
| | | 2006 | 7205 | 24.7722 | 20 | 0.068764 | 21.0211 | 226600 | ... |
| | | 2012 | 7205 | 24.7722 | 20 | 0.068764 | 21.0211 | 227000 | ... |
| | 2 | 2000 | 21824 | 75.0352 | 4 | 0.0137528 | 74.3614 | 223900 | ... |
| | | 2006 | 21880 | 75.2278 | 4 | 0.0137528 | 74.5539 | 226600 | ... |
| | | 2012 | 21880 | 75.2278 | 4 | 0.0137528 | 74.5539 | 227000 | ... |
| 15000 | 1 | 2000 | 9630 | 16.7106 | 46 | 0.0798223 | 11.5326 | 395200 | ... |
| | | 2006 | 9278 | 16.0998 | 49 | 0.0850281 | 11.2671 | 391300 | ... |
| | | 2012 | 9320 | 16.1727 | 50 | 0.0867634 | 11.2671 | 395500 | ... |
| | 2 | 2000 | 47998 | 83.2894 | 4 | 0.00694107 | 82.9493 | 395200 | ... |
| | | 2006 | 48350 | 83.9002 | 4 | 0.00694107 | 83.5601 | 391300 | ... |
| | | 2012 | 48308 | 83.8273 | 4 | 0.00694107 | 83.4872 | 395500 | ... |
| 20000 | 1 | 2000 | 12149 | 13.3476 | 76 | 0.0834981 | 7.30169 | 565200 | ... |
| | | 2006 | 11827 | 12.9938 | 78 | 0.0856955 | 7.1336 | 566200 | ... |
| | | 2012 | 11882 | 13.0543 | 79 | 0.0867941 | 7.1336 | 571400 | ... |
| | 2 | 2000 | 78871 | 86.6524 | 5 | 0.0054933 | 86.3151 | 565200 | ... |
| | | 2006 | 79193 | 87.0062 | 6 | 0.00659196 | 86.6678 | 566200 | ... |
| | | 2012 | 79138 | 86.9457 | 7 | 0.00769062 | 86.604 | 571400 | ... |

will plot the temporal evolution of the area-weighted fractal dimension at the landscape level for the three buffer distances in the same axis, producing an output as depicted in Fig 10.

Although this is beyond the scope of this article, the above plot suggests that the area-weighted fractal dimension shows a predictable response to changing the spatial extent of the considered landscape [16, 17].

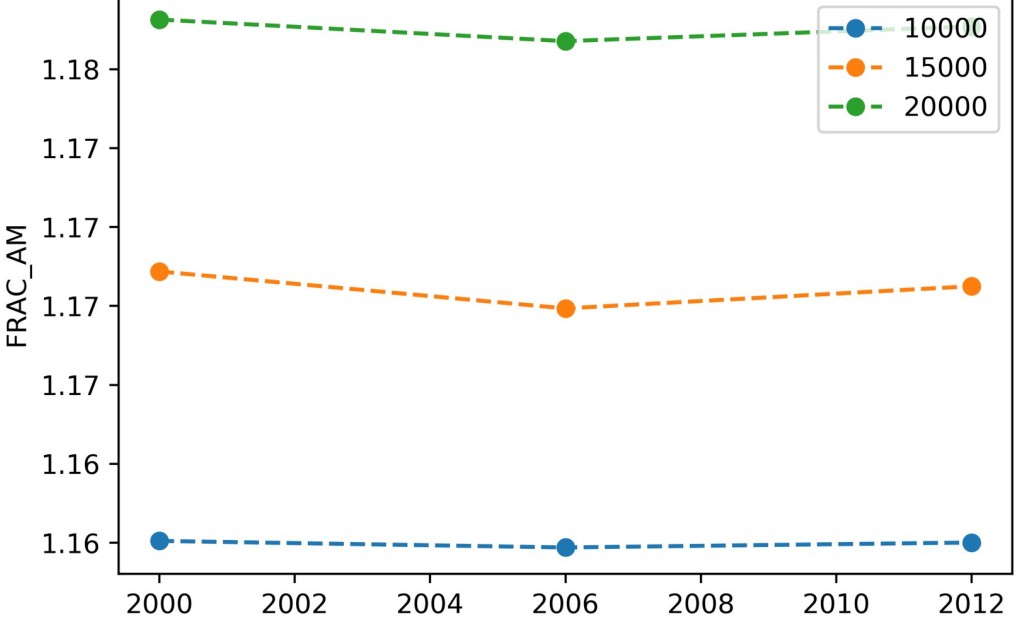

**Fig 10. Example of a plot for a landscape-level metric in a spatiotemporal buffer analysis.**

## The PyLandStats library

### Availability and installation

The source code of PyLandStats is available in a GitHub repository at https://github.com/martibosch/pylandstats, and is licensed under the open source GNU General Public License 3 (GNU GPLv3) to ensure that any derivative work is kept as open source. The easiest way to install PyLandStats is by installing the dedicated conda recipe hosted on the conda-forge channel at https://anaconda.org/conda-forge/pylandstats, as in:

```
$ conda install -c conda-forge pylandstats
```

The above command will install all the necessary requirements to run all the features of PyLandStats. Alternatively, a dedicated Python package is hosted on the Python Package Index (PyPI) at https://pypi.org/project/pylandstats/, and can be readily installed with `pip` as in:

```
$ pip install pylandstats
```

Nevertheless, the `BufferAnalysis` and `SpatioTemporalBufferAnalysis` classes have dependencies that cannot be installed with `pip`, namely the Geospatial Data Abstraction Library (GDAL) and the Geometry Engine Open Source (GEOS). In order to use these two PyLandStats classes, GDAL and GEOS must be present at the time of installing PyLandStats, which in this case will further require specifying the `geo` extra requirements as in:

```
$ pip install pylandstats[geo]
```

Unit tests are run within the Travis Continuous Integration (Travis CI) platform at https://travis-ci.org/martibosch/pylandstats every time that new commits ar pushed to the GitHub repository. Additionally, test coverage is reported on Coveralls at https://coveralls.io/github/martibosch/pylandstats?branch=master.

The documentation of PyLandStats is hosted in Read the Docs at https://pylandstats.readthedocs.io/ and is also available in S1 Text. Additionally, a collection of example notebooks with a thorough overview of PyLandStats's features is provided at a dedicated GitHub repository at https://github.com/martibosch/pylandstats-notebooks, which can be executed interactively online by means of the Binder web service [18]. Such repository includes unit tests which ensure the correctness of the computations (see S5 Code).

Finally, an example application of PyLandStats in an academic article can be found in the analysis of the spatiotemporal patterns of urbanization of three Swiss urban agglomerations by Bosch and Chenal [19], and all the code and materials necessary to reproduce the results are available in a dedicated GitHub repository at https://github.com/martibosch/swiss-urbanization.

### Dependencies and implementation details

The PyLandStats package is fully implemented in Python, and requires the Python packages NumPy, SciPy, pandas, matplotlib, rasterio. The first four are among the most popular packages for scientific and data-centric Python and are used for a wide-variety of scientific needs, whereas rasterio is a popular library to read and write geospatial raster data. In PyLandStats, NumPy arrays are used to represent landscapes and patch-level metrics. In addition, NumPy functions are used in the computations of all the implemented landscape metrics. The SciPy library is used to segment the patches in the landscape arrays, compute the inter-patch nearest-neighbor distances, and to compute the coefficient of variation of the patch-level landscape metrics. The pandas data frames are used to build the data frames of landscape metrics, matplotlib is used to produce the plots and rasterio is used to read raster data, plot the landscapes as well as to rasterize the vector geometries used in `BufferAnalysis` and `SpatioTemporalBufferAnalysis`. As noted above, the foregoing two classes further require the GeoPandas and Shapely Python packages.

The implementation of PyLandStats is organized in Python modules, where the classes described throughout this paper are defined. Such object-oriented design offers many advantages. On the one hand, it allows both for a conceptual separation and reusability of the functionalities, which enhances the maintainability and extensibility of PyLandStats. On the other hand, Python properties serve to cache results that are computationally expensive to obtain, which can later be accessed in constant (almost immediate) time. This mechanism is exploited to cache intermediate results that are later used to compute the metrics (see S6 Code). More precisely, instances of the `Landscape` class cache the list of patches, each with its respective LULC class, area, perimeter and nearest-neigbhor distance, as well as the pixel adjacency matrix, i.e., the number of adjacencies between pixels of each landscape class (including adjacencies between pixels of the same class). Furthermore, such mechanism eases the task of implementing new metrics, since the vast majority of landscape metrics found throughout the academic literature can be straight-forwardly computed out of such cached properties (see the section 1.1 "List of implemented metrics" of S1 Text as well as [5]). Finally, as follows from the cache mechanism described above, the memory size of a `Landscape` instance scales linearly with the number of patches present in the respective raster landscape.

Regarding the performance, the most expensive operations of PyLandStats are the computation of the adjacency matrix, and more importantly, the computation of the inter-patch nearest-neighbor distances. The code for the former is transformed from Python to C++ by means of the Pythran ahead-of-time compiler [20], which achieves speed-ups of an order of magnitude of three. The code for the latter consists of a slow nested Python loop that iterates over each patch of each class and employs SciPy's implementation of the K-d tree in Cython [21] in order to find the nearest neigbor of each patch. The computation of the inter-patch nearest-neighbor distances is by far the main performance bottleneck of PyLandStats (see S6 Code), and it is therefore recommended that in analysis cases that do not require computing euclidean nearest-neighbor metrics avoid its computation by making use of the `metrics` keyword argument as explained above.

## Improvements of PyLandStats over existing software packages

There have been many other freely-available software packages to compute landscape metrics [6] (see Table 12). By far, the most popular one has been FRAGSTATS [22], yet as a stand-alone software, its functions cannot be directly integrated into advanced computational workflows. Furthermore, FRAGSTATS is not open-source software. Recently, the open-source R package landscapemetrics [23] has been developed to overcome such shortcomings by relying on a well-

**Table 12. Comparison of FRAGSTATS, landscapemetrics, LecoS and PyLandStats.**

| Characteristic | FRAGSTATS | landscapemetrics | LecoS | PyLandStats |
|---|---|---|---|---|
| open source | no | yes | yes | yes |
| programming language | ? | R | Python | Python |
| cross-platform compatibility | no | yes | yes | yes |
| integration into advanced workflows | no | yes | QGIS only | yes |
| Benchmark Vaud [s] | 0.61 | 14.27 | - | 0.91 |
| Benchmark Bern and Valais [s] | 33.31 | 553.45 | - | 32.2 |

The two benchmarks consist in the computation of the 95 metrics implemented in PyLandStats for the landscape snapshots of the canton of Vaud (889x916 pixels of 2 LULC classes) and the cantons of Bern and Valais (1640x1319 pixels of 28 LULC classes) respectively (see S6 Code for more details). Both landscapes have been derived from the Corine Land Cover [7] dataset for the year 2000. Note that LecoS has been excluded from the benchmarks since only features 20 landscape metrics.

established spatial framework in R. On the other hand, the only available tool to compute landscape metrics in Python is the LecoS package [24], which is designed as a QGIS plugin.

The computed values for the landscape metrics in PyLandStats are the same as in FRAG-STATS, with a maximum relative difference of 0.1% (see S5 Code). Furthermore, the performance of both packages is very similar. Nevertheless, unlike FRAGSTATS, PyLandStats is open source and it is therefore straightfoward for users to contribute to its development on its GitHub repository. On the other hand, PyLandStats is an alternative to landscapemetrics for those users that prefer to write their computational workflows in Python rather than R. Additionally, the cache mechanisms included within PyLandStats lead to significantly better performance and make it more suitable for experimentation in interactive environments such as Jupyter notebooks [25], since it ensures that the marginal cost of subsequent calls to compute a metric are minimal (see S6 Code).

Finally, although LecoS is based on the NumPy and SciPy stack (like PyLandStats), only 20 metrics have been implemented, and its design as a QGIS plugin forces the users to adapt the computational workflows to QGIS. In sharp contrast, PyLandStats is designed as a Python package which can be directly used in Python scripts, Jupyter notebooks and in other Python packages including QGIS plugins.

In view of the growing popularity of Jupyter notebooks and continuous releases of new Python packages to visualize geospatial data interactively, it is reasonable to expect that geospatial scientists, including landscape ecologists, will increasingly turn to the Jupyter environments for their analyses. From this perspective, PyLandStats intends to offer a Python package that geospatial scientists can use in order to compute landscape metrics, and whose modularity and object-oriented design allows it to evolve and adapt to new developments in the Python and Jupyter ecosystem.

## Supporting information

**S1 Text. PyLandStats documentation.**
(PDF)

**S1 Table. Table of metrics implemented in PyLandStats.**
(PDF)

**S1 Code. Landscape analysis with PyLandStats for the canton of Vaud (Switzerland), as Jupyter notebook.**
(IPYNB)

**S2 Code. Spatiotemporal analysis with PyLandStats for the canton of Vaud (Switzerland), as Jupyter notebook.**
(IPYNB)

**S3 Code. Zonal analysis with PyLandStats for the canton of Vaud (Switzerland), as Jupyter notebook.**
(IPYNB)

**S4 Code. Spatiotemporal buffer analysis with PyLandStats for the canton of Vaud (Switzerland), as Jupyter notebook.**
(IPYNB)

**S5 Code. Comparison of the metrics computed in FRAGSTATS v4 and PyLandStats for the canton of Vaud (Switzerland), as Jupyter notebook.**
(IPYNB)

**S6 Code. Performance notes and benchmarks comparing FRAGSTATS v4, landscape-metrics and PyLandStats, as Jupyter notebook.**
(IPYNB)

## Author Contributions

**Conceptualization:** Martí Bosch.

**Data curation:** Martí Bosch.

**Software:** Martí Bosch.

**Visualization:** Martí Bosch.

**Writing – original draft:** Martí Bosch.

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
