## [Decision Letter · Decision Letter 0]

13 Sep 2019

PONE-D-19-21282

PyLandStats: An open-source Pythonic library to compute landscape metrics

PLOS ONE

Dear %TITLE% Bosch,

Thank you for submitting your manuscript to PLOS ONE. After careful consideration, we feel that it has merit but does not fully meet PLOS ONE’s publication criteria as it currently stands. Therefore, we invite you to submit a revised version of the manuscript that addresses the points raised during the review process.

We would appreciate receiving your revised manuscript by Oct 28 2019 11:59PM. To enhance the reproducibility of your results, we recommend that if applicable you deposit your laboratory protocols in protocols.io, where a protocol can be assigned its own identifier (DOI) such that it can be cited independently in the future. For instructions see: http://journals.plos.org/plosone/s/submission-guidelines#loc-laboratory-protocols

We look forward to receiving your revised manuscript.

Kind regards,

Prof. Duccio Rocchini, PhD

Academic Editor

PLOS ONE

Journal requirements;

1. Thank you fo rincluding the following funding information within your acknowledgements section; "This research has been supported by the Ecole Polytechnique Federale de Lausanne 397

(EPFL). The Corine Land Cover inventory used for the example data were produced 398

with funding by the European Union."

Reviewers' comments:

Reviewer's Responses to Questions

**Comments to the Author**

1. Is the manuscript technically sound, and do the data support the conclusions?

Reviewer #1: Yes

Reviewer #2: Yes

2. Has the statistical analysis been performed appropriately and rigorously? 

Reviewer #1: N/A

Reviewer #2: Yes

3. Have the authors made all data underlying the findings in their manuscript fully available?

Reviewer #1: Yes

Reviewer #2: Yes

4. Is the manuscript presented in an intelligible fashion and written in standard English?

Reviewer #1: Yes

Reviewer #2: Yes

5. Review Comments to the Author

Reviewer #1: This article introduces PyLandStats, a definitely needed Python-based alternative to other patch metrics packages.

General comments:

* I would prefer having the patch metrics listed in the article rather than refer to the documentation, it seems like important information readers would want to see right away. How were these metrics selected? (around line 70)

* I miss benchmark to see how much memory and time is required. What are the limitations, I assume everything is in memory? It could be also useful to see how demanding are individual metrics.

* Did you compare the results with other packages? Do the automated tests actually test the correctness of the computations (e.g. compare with manually verified values)?

* The quality of the figures is really bad, I wonder if the submission system messed that up?

Software comments:

* why there are functions for computing metrics for Landscape, but only properties for spatio-temporal and buffer analysis? Seems to be inconsistent, if possible comment on that.

* how are dates specified, could it be e.g. DateTime object as well?

* you call it gradient analysis, however, especially the generic case sounds more like zonal analysis in GIS terms, perhaps worth clarifying that in the manuscript.

* L152: might be better to mention matplotlib as the library used for plotting in the beginning of the section

* L112: perhaps clarify when the metrics are computed, if I understood correctly, it's not during the initialization but once you get the property.

Typos:

L20: missing 'and'

L125: the the

L138: skip 'probably'

L146: slug?

Figure 1 caption: metric

L159: consists of

L192: across

L195: missing space

L196: I would add space between number and unit

L203: weird formulation

L211: skip 'really'

L263: scope of this article

L268: in a GitHub repository

L276: dependencies

L282: are

L300: neighbor

L300: inter-patch (or one word), see also other places

L324: of adjacency matrix

L335: software packages

L368:object-oriented what?

in references there are multiple places where it needs upper case in titles

Reviewer #2: In this study the author introduce PyLandStats, a new Python library to compute and quantify landscapes spatial patterns.

I’ve found the paper interesting and the library really useful, especially because before the recent Hesselbarth et al. (2019) there were not (to the best of my knowledge) R packages or Python libraries to efficiently compute a whole set of landscape statistics without relying to standalone GIS software.

The figures provided have a low quality, I recommend the author to improve them. The palette of Fig 3,5,7,8 could also be improved to increase the readability of the figure.

It would be nice to see some comparison between the proposed packaged and the R pkg landscapemetrics, as well as for FRAGSTATS and LecoS QGIS plugin, in term of both accuracy and precision of the measures and time computation.

I cannot access to the GitHub repository, something went wrong with the link?

Personal curiosity: the library relies on rasterio, have you found some issues regarding the incompatibility between rasterio and gdal python libraries ? (e.g. https://rasterio.readthedocs.io/en/stable/topics/switch.html).

6. PLOS authors have the option to publish the peer review history of their article (what does this mean?). If published, this will include your full peer review and any attached files.

Reviewer #1: No

Reviewer #2: No

---

## [Author Response · Author response to Decision Letter 0]

1 Oct 2019

# Rebuttal letter

## Editor

In order to avoid any misunderstandings, the "Acknowledgements" section has been dropped. The funding statement can be left as it is since the authors have not received any specific funding for this work.

## Reviewer #1

The metrics have been listed in Table S1. The choice of implemented metrics has been mostly guided by their recurrence in the studies of land use/land cover change associated to urbanization. As detailed in the section "The PyLandStats library", the organization of the code and its open-source model enable users to contribute to the repository by requesting the implementation of new metrics or actually implementing them. On the other hand, lines 68-74 do not intend to suggest which metrics should be selected, but instead to show that users might employ the metrics keyword argument in order to limit their analysis to a subset of metrics of their choice.

Two benchmarks comparing the execution times with those of FRAGSTATS and landscapemetrics has been included as Code S6. Such supporting information also features further details regarding the performance of PyLandStats. Additionally, a note regarding the memory size of Landscape objects has been added to the section "Dependencies and implementation details"

The values computed for all the metrics implemented in PyLandStats are compared with those computed with FRAGSTATS v4 in Code S5, showing a maximum discrepancy of 0.1%. Such notebook is included in the example notebook repository (https://github.com/martibosch/pylandstats-notebooks), which is also subject to automated tests in Travis CI. Therefore, the automated tests for such repository do test for the correctness of the computations.

I have updated the figures with higher DPI and updated the color palette of the landscape raster plots.

Properties were used to compute the data frames of metrics in spatiotemporal, buffer and gradient analysis in order to avoid recomputing the metrics each time that the user desired to obtain a data frame or plot. Nevertheless, after the in-depth exploration of the performance detailed in Code S6, it seems that such performance gains are negligible. Therefore, in order to ensure a consistent API, the computation of data frames of metrics will use methods (instead of properties) throughout all the classes of PyLandStats.

I have added a note (and updated the notebook in Code S2) to acknowledge that dates might also be specified as strings or datetime objects.

The generic GradientAnalysis class has been renamed to ZonalAnalysis, and the manuscript and notebook of Code S3 have updated accordingly.

A reference to matplotlib has been added (with its corresponding citation). For consistency, the citations for numpy and pandas have also been added (note that scipy, rasterio, shapely and geopandas do not have proper academic articles).

I believe that after the unification of the API regarding the computation of data frames with methods, it is no longer necessary to clarify when metrics are computed since they are indeed computed when the methods are called. Additionally, details regarding how metrics are computed in PyLandStats are covered in section "Dependencies and implementation details" as well as Code S6.

I have corrected the typos.

## Reviewer #2

I have updated the figures with higher DPI and updated the color palette of the landscape raster plots.

Two benchmarks comparing the execution times with those of FRAGSTATS and landscapemetrics has been included as Code S6. Such supporting information also features further details regarding the performance of PyLandStats.

The values computed for all the metrics implemented in PyLandStats are compared with those computed with FRAGSTATS v4 in Code S5, showing a maximum discrepancy of 0.1%.

The link to the GitHub repository is displayed correctly in the manuscript (https://github.com/martibosch/pylandstats) and I believe that it has been working all along.

Regarding the incompatibility between rasterio and GDAL's Python bindings, I have never encountered any issues, neither when installing PyLandStats via pip nor when installing it via conda. In any case, users might report any of such eventual issues in the GitHub repository.

---

## [Decision Letter · Decision Letter 1]

12 Nov 2019

PyLandStats: An open-source Pythonic library to compute landscape metrics

PONE-D-19-21282R1

Dear Dr. Bosch,

We are pleased to inform you that your manuscript has been judged scientifically suitable for publication and will be formally accepted for publication once it complies with all outstanding technical requirements.

With kind regards,

Duccio Rocchini, PhD

Academic Editor

PLOS ONE

Additional Editor Comments (optional):

Reviewers' comments:

Reviewer's Responses to Questions

**Comments to the Author**

1. If the authors have adequately addressed your comments raised in a previous round of review and you feel that this manuscript is now acceptable for publication, you may indicate that here to bypass the “Comments to the Author” section, enter your conflict of interest statement in the “Confidential to Editor” section, and submit your "Accept" recommendation.

Reviewer #1: (No Response)

Reviewer #2: All comments have been addressed

2. Is the manuscript technically sound, and do the data support the conclusions?

Reviewer #1: Yes

Reviewer #2: Yes

3. Has the statistical analysis been performed appropriately and rigorously? 

Reviewer #1: Yes

Reviewer #2: Yes

4. Have the authors made all data underlying the findings in their manuscript fully available?

Reviewer #1: Yes

Reviewer #2: Yes

5. Is the manuscript presented in an intelligible fashion and written in standard English?

Reviewer #1: Yes

Reviewer #2: Yes

6. Review Comments to the Author

Reviewer #1: Thank you for addressing my comments and putting the effort into reproducible research. I am looking forward to using the library. Found just couple small typos, which could be fixed:

L119: Likewise Landscape instances ('Similarly to' would be better)

L154: follow -> following

L330: is scales

Reviewer #2: the author has answered to all the comments and provided an updated version of the manuscript. I would recommend to accept the paper. Well done!

7. PLOS authors have the option to publish the peer review history of their article (what does this mean?). If published, this will include your full peer review and any attached files.

Reviewer #1: No

Reviewer #2: No

---

## [Editor Report · Acceptance letter]

20 Nov 2019

PONE-D-19-21282R1 

PyLandStats: An open-source Pythonic library to compute landscape metrics 

Dear Dr. Bosch:

I am pleased to inform you that your manuscript has been deemed suitable for publication in PLOS ONE. Congratulations! Your manuscript is now with our production department. 

With kind regards,

on behalf of

Dr. Duccio Rocchini 

Academic Editor

PLOS ONE